# Psychological therapies for adolescents with borderline personality disorder (BPD) or BPD features—A systematic review of randomized clinical trials with meta-analysis and Trial Sequential Analysis

**Mie Sedoc Jørgensen**[1,2,3]\*, **Ole Jakob Storebø**[1,2,4], **Jutta M. Stoffers-Winterling**[5], **Erlend Faltinsen**[1], **Adnan Todorovac**[1], **Erik Simonsen**[1,3]

**1** Psychiatric Research Unit, Region Zealand, Denmark, **2** Child and Adolescent Psychiatric Department, Region Zealand, Denmark, **3** Faculty of Health and Medical Sciences, Department of Clinical Medicine, University of Copenhagen, Copenhagen, Denmark, **4** Department of Psychology, University of Southern Denmark, Zealand, Denmark, **5** Department of Psychiatry and Psychotherapy, University Medical Center Mainz, Mainz, Germany

\* mipjo@regionsjaelland.dk

## Abstract

### Objectives

To review the effectiveness of psychological therapies for adolescents with borderline personality disorder (BPD) or BPD features.

### Methods

We included randomized clinical trials on psychological therapies for adolescents with BPD and BPD features. Data were extracted and assessed for quality according to Cochrane guidelines, and summarized as mean difference (MD) with 95% confidence intervals (CI) for continuous data and as Odds ratios (OR) with 95% CI for dichotomous data. Risk of bias was assessed using Cochrane's risk of bias tool for each domain. When possible, we pooled trials into meta-analyses, and used Trial Sequential Analysis (TSA) to control for random errors. Quality of the evidence was assessed using the Grading of Recommendations, Assessment, Development, and Evaluation (GRADE).

### Results

10 trials on adolescents with BPD or BPD features were included. All trials were considered at high risk of bias, and the quality of the evidence was rated as "very low". We did TSA on the primary outcome and found that the required information size was reached. The risk of random error was thus discarded.

### Conclusion

Only 10 trials have been conducted on adolescents with BPD or BPD features. Of these only few showed superior outcomes of the experimental intervention compared to the

**Data Availability Statement:** All relevant data are within the manuscript and its Supporting information files.

**Funding:** TrygFonden (grant number 115638), Psychiatric Research Unit Region Zealand, the Health Scientific Research Fund of Region Zealand, and Department of Health and Medical Sciences, University of Copenhagen.

**Competing interests:** The authors have read the journal's policy and have the following competing interests: MSJ is associated with the M-GAB trial and trained in Dialectical behavior therapy and psychodynamic therapy. OJS is associated with the M-GAB trial, trained in child and adolescent psychoanalytic play therapy and trained in group psychoanalysis. JSW is a board-certified behavior therapist and trained in Dialectical behavior therapy. ES is associated with the M-GAB trial and trained in group psychoanalysis. This does not alter our adherence to PLOS ONE policies on sharing data and materials.

control intervention. No adverse effects of the interventions were mentioned. Attrition rates varied from 15–75% in experimental interventions. The overall quality was very low due to high risk of bias, imprecision and inconsistency, which limits the confidence in effect estimates. Due to the high risk of bias, high attrition rates and underpowered studies in this area, it is difficult to derive any conclusions on the efficacy of psychological therapies for BPD in adolescence. There is a need for more high quality trials with larger samples to identify effective psychological therapies for this specific age group with BPD or BPD features.

## Introduction

Borderline personality disorder (BPD) is a severe mental disorder characterized by a pervasive pattern of instability in affect, impulse control, interpersonal relationships and behavior [1,2]. BPD manifests during childhood or adolescence, and we now know that BPD can be validly diagnosed in adolescence [3–5]. In spite of this, many clinicians have been reluctant in diagnosing personality disorders in youth. This delay in diagnosing personality disorders in youth heightens the risk of ineffective or even iatrogenic treatments (psychotherapeutic and/or psycho-pharmacological), and could be a risk factor of decreases in psychosocial functioning over time [4,6]. In order to take action to prevent the development of this disorder, an increasing amount of randomized clinical trials (RCTs) have been conducted on early detection and treatments for adolescents with BPD or BPD features.

Reviews on psychological therapies for BPD have focused on patients who fulfilled diagnostic threshold for BPD and therefore have excluded patients who presented with BPD features (at a subthreshold level). This means that clinicians and researchers are left uninformed about the evidence base and the quality of evidence of RCTs on psychological therapies for BPD pathology in adolescence. The aim of this review was thus to assess the effectiveness of psychological therapies for adolescents with BPD or BPD features.

## Methods

This review was conducted according to a published protocol [7] in close collaboration with the Cochrane review of psychological therapies for BPD, and therefore the method section will resemble that seen in the Cochrane review [8], but the participants, interventions, comparisons and outcomes will be different. We will present additional data that was excluded from the Cochrane review on psychological therapies for BPD (due to the requirement of a full criteria BPD diagnosis) with analyses that focus on trials for adolescents with BPD or BPD features.

### Study selection

We considered RCTs of psychotherapeutic treatments for adolescents with BPD or BPD features eligible for inclusion. Trials were included irrespective of language, and publication year, type or status.

### Types of participants

Patients were eligible if they had a formal diagnosis of BPD according to the *Diagnostic of Statistical Manual of Mental Disorders* (DSM), Third Edition (DSM-III) [9], Third Edition Revised (DSM-III-R) [10], Fourth Edition (DSM-IV) [11], Fourth Edition Text Revision (DSM-IV-TR) [12], and Fifth Edition (DSM-5) [1], and also if they presented with BPD

features at any level (i.e. any trial that specifically targeted BPD symptoms at a threshold or subthreshold level as an overall aim of the trial).

We included trials involving subsamples of BPD patients providing data on these patients were available separately. We included adolescent participants with BPD or BPD features, with or without any comorbid psychiatric conditions. We excluded trials that focused on mental impairment, organic brain disorder, dementia or other severe neurologic/neurodevelopmental diseases [7].

## Types of interventions

This review included the same types of experimental and comparator interventions as the Cochrane review [7,8]. Experimental interventions included any well-defined, theory-driven psychotherapeutic treatment. We included all types of psychotherapy, regardless of theoretical orientation or treatment setting. We included any kind of treatment setting: inpatient, outpatient or partially hospitalized. We included the following types of interventions: 1) individual psychotherapy, 2) group psychotherapy, 3) family therapy and 4) any combination of individual, family and/or group psychotherapy.

We included the following comparator interventions: 1) control interventions (e.g., standard care, treatment as usual [TAU] or waitlist or no treatment), and 2) specific psychotherapeutic interventions that were well-defined and theory-driven, e.g., general psychiatric management [13]. Concomitant treatments were included if they were applied to both treatment conditions.

## Types of outcome measures

Outcomes were self-rated by patients or observer-rated by clinicians.

We analyzed all outcomes at end of treatment (EOT) and at any potential follow-up periods.

If a trial included several measures of the same outcome, we only included one instrument per outcome to avoid double counting of participants. We preferred the observer-rated instruments over the self-rated.

**Primary outcome.**   For the primary outcome we chose *BPD severity*. This includes any measure of BPD symptoms, features or severity.

## Secondary outcomes

1. *Psychopathological syndromes* included all psychopathological syndromes (except BPD) such as depression, anxiety, psychoticism etc. If the trials included more than one measure of psychopathological syndromes, we included the most morbid syndrome for this review (for example, psychotic disorders over depression, but depression over anxiety).

2. *Impulsivity* covered self-harm, non-suicidal self-injury (NSSI), suicide attempts and externalizing behaviors

3. *Substance or alcohol abuse* included substance abuse or dependence (all substances) or alcohol abuse or dependence.

4. *Functioning* included global functioning, occupational functioning and interpersonal functioning.

5. *Quality of life* included all measures on quality of life.

6. *Attrition* included number of patients lost after randomization in each intervention due to any reason

7. *Adverse events* included unfavorable outcomes that occur during or after psychological therapies but not necessarily caused by it. Adverse events are divided into *severe adverse events* (any event that leads to death, is life-threatening, requires inpatient hospitalization or prolongation of existing hospitalization, results in persistent or significant disability, and any medical even that may have jeopardized the participant's health or requires intervention to prevent it), and *non-serious adverse events* (all other adverse events) [8].

## Search methods for identification of trials

We searched in the electronic databases and trial registers listed below:

1. Cochrane Central Register of Controlled Trials (CENTRAL; current issue), in the Cochrane Library, which includes the Cochrane Developmental, Psychosocial and Learning Problems Specialised Register

2. MEDLINE Ovid (1948 onwards)

3. Embase Ovid (1980 onwards)

4. CINAHL EBSCOhost (Cumulative Index to Nursing and Allied Health Literature; 1980 onwards)

5. PsycINFO Ovid (1806 onwards)

6. ERIC EBSCOhost (Education Resources Information Center; 1966 onwards)

7. BIOSIS Previews Web of Science Clarivate Analytics (1969 onwards)

8. Web of Science Core Collection Clarivate Analytics (1900 onwards)

9. Sociological Abstracts ProQuest (1952 onwards)

10. LILACS (Latin American and Caribbean Health Science Information database; lilacs.bvsalud.org/en)

11. OpenGrey (www.opengrey.eu)

12. Copac National, Academic and Specialist Library Catalogue (COPAC; copac.jisc.ac.uk)

13. ProQuest Dissertations and Theses A&I (1743 onwards)

14. DART Europe E-Theses Portal (www.dart-europe.eu/basicsearch.php)

15. Networked Digital Library of Theses and Dissertations (NDLTD; www.ndltd.org)

16. Australian New Zealand Clinical Trials Registry (ANZCTR; www.anzctr.org.au/BasicSearch.aspx)

17. ClinicalTrials.gov (clinicaltrials.gov)

18. EU Clinical Trials Register (www.clinicaltrialsregister.eu/ctr-search/search)

19. ISRCTN Registry (www.isrctn.com)

20. UK Clinical Trials Gateway (www.ukctg.nihr.ac.uk/#popoverSearchDivId)

21. WHO International Clinical Trials Registry Platform (ICTRP; who.int/ictrp/en)

In depth details on the data sources and search criteria are available in the Cochrane systematic review [7,8].

## Data collection and analysis

We made this review according to the *Cochrane Handbook for Systematic reviews of Interventions* [14], and analyses were performed using Review Manager 5.3 (RevMan)—the Cochrane Collaboration's statistical software.

## Data extraction and quality assessment

Data were extracted as a part of the Cochrane Review [7,8]. Six review authors independently extracted data for this review and assessed risk of bias according to Cochrane's tool for assessing risk of bias [14]. The following risk of bias domains were assessed and subsequently determined in pairs of two data extractors: random sequence generation, allocation concealment, blinding of participants and personnel, blinding of outcome assessment, incomplete outcome data, selective outcome reporting, and other sources of bias. For each domain, the data extractors assigned the included trials to one of three categories (low risk of bias, unclear risk of bias or high risk of bias), according to guidelines provided in the *Cochrane Handbook for Systematic Reviews of Interventions* [14]. We considered trials with one or more unclear or high risk of bias domains as high risk of bias trials. We resolved disagreements by discussion or use of an arbiter if needed. Trial authors were contacted in case we needed supplementary data or information.

We assessed and graded the quality of the evidence according to the Grading of Recommendations, Assessment, Development, and Evaluation (GRADE). The GRADE approach was used to construct a 'Summary of findings' table where all review outcomes were presented. The GRADE approach evaluates the quality of a body of evidence in terms of risk of bias, directness of the evidence, heterogeneity of the data, precision of effect estimates, and risk of publication bias [15]. The overall GRADE-evaluation indicates to which extent one can be confident in the effect estimates.

## Data synthesis and statistical analysis

For continuous data, we compared the mean score between the two groups to give a mean difference (MD). This is presented with 95% confidence intervals (CIs). When there were different outcome measures used on the same construct, we estimated the standardized MD (SMD). We calculated standardized mean differences (SMDs) using end-scores on the basis of end of treatment results and follow-up data, respectively. If the trials did not report means and standard deviations but reported other values such as t-tests and P values, we tried to transform these into standard deviations. Dichotomous data were summarized as Odds ratios (OR) with 95% CIs. We calculated study estimates on the basis of end of treatment results and did separate analyses for different therapies as well as for follow-up data. Whenever there were incomplete reports or missing data on outcomes stated as having been assessed, we contacted the trial authors.

We performed the statistical analysis according to recommendations in the latest version of the *Cochrane Handbook for Systematic Reviews of Interventions* [16]. Meta-analyses were carried out even if there was concern about heterogeneity. If the heterogeneity was very high in the meta-analysis, we downgraded the quality by using the GRADE tool [15]. When carrying out the meta-analysis, we used the inverse variance method to give more precise estimates from trials with less variance (mostly larger trials) more weight. We used the random-effects model for meta-analysis since we expected some degree of clinical heterogeneity to be present

in most cases and the fixed effect model when presenting singe trial results. If data pooling seemed feasible, we pooled the primary trials effects and calculated their 95% CIs.

### Diversity-adjusted required information size and Trial Sequential Analysis

Trial Sequential Analysis (TSA) is a methodology that combines a required information size (RIS) calculation for a meta-analysis with the threshold for statistical significance [17–19]. TSA is used to quantify the statistical reliability of the data in cumulative meta-analysis adjusting P values for sparse data and for repetitive testing on accumulating data thus controlling the risks of type I and type II errors [17–19]. We calculated the diversity-adjusted required information size (DARIS, i.e. number of participants required to detect or reject effects in meta-analyses), and used TSA for our primary outcome at end of treatment. If the TSA did not find a significant finding before the RIS was reached (no Z curve crossing of the trial sequential monitoring boundaries), we could conclude that either more trials were needed to reject or accept an intervention effect or the anticipated effect could be rejected. If the cumulated Z curve enters the futility area, the anticipated intervention effect can be rejected.

## Results

### Results of the search

We carried out electronic searches over three time periods and this review fully incorporates the results of searches conducted until March 2019. The PRISMA flow chart (Fig 1) shows the trials identified, screened, and included for this review. We identified ten trials that consisted of adolescent samples with BPD or BPD features. Of these ten trials, six were included in the Cochrane review of psychological therapies for BPD, but only with data on participants who met diagnostic criteria for BPD [8]. Three trials had been excluded from the Cochrane review because either less than 70% of the total sample had full threshold BPD or because no subsample data were delivered on participants with BPD. However, these three trials all met inclusion

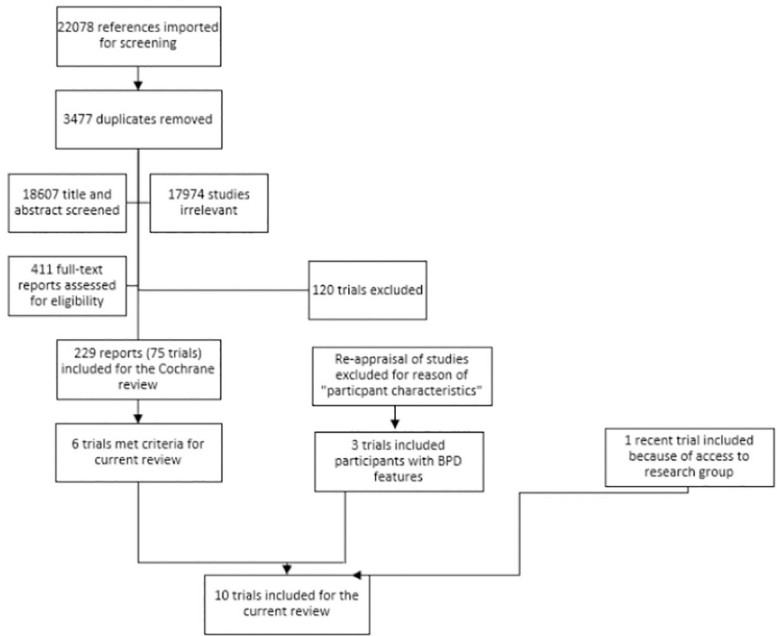

**Fig 1. PRISMA flow chart.**

criteria for the current review, as the re-inspection of all excludes for the reason of participant characteristics showed. One trial on mentalization-based group treatment [20,21] was published shortly after the last search had been carried out, but we were able to include this trial because MSJ was a part of the research group. The decision about trial inclusion, however, as well as risk of bias assessment and data analyses of this trial were done by other authors not affiliated with the trial (EGF and AT).

## Description of included trials

We included a total of 10 randomized clinical trials involving adolescents with BPD or BPD features for this review. The total number of participants were 775. Only six trials reported having done a sample size calculation before the start of the trial [20,22–26]. In nine of the trials, we included the entire sample of participants, because they all met the criteria for either BPD or had BPD features [20,22–24,26–30]. In one trial, only 59.1% of the sample had BPD (the remaining fulfilled criteria of a mixed disorder of conduct and emotions) [25], but subsample data for the BPD participants was already included into the Cochrane review [8]. For a detailed overview of the included trials, please see Table 1.

**Table 1. Description of included trials.**

| Trial | Country | Setting | *n* | Age | No. of BPD criteria | Experimental | Control | Duration |
|---|---|---|---|---|---|---|---|---|
| Chanen et al., 2008 [22] | Australia | Outpatient | *n* = 86 (76% female) | 16.4 (*SD* = 0.9) | 41% BPD, 59% ≥ 2 | CAT | SGCC | 24 weeks |
| Schuppert et al., 2009 [31] | The Netherlands | Outpatient | *n* = 43 (88.4% female) | 16.14 (*SD* = 1.23) | ns. ≥ 2 | ERT + TAU | TAU | 17 weeks plus two booster sessions, control ns. |
| Schuppert et al., 2012 [27] | The Netherlands | Outpatient | *n* = 109 (96% female) | 15.98 (*SD* = 1.22) | 73% BPD, 27% ≥ 2 | ERT + TAU | TAU | 17 weeks plus two booster sessions, control ns. |
| Gleeson et al., 2012 [28] | Australia | Outpatient | *n* = 16 (81.2% female) | 18.4 (*SD* = 2.9) | 75% BPD, 25% ≥ 4 | CAT + SFET | SFET | CAT for 16 weeks, EOT at 6 months |
| Rossouw & Fonagy, 2012 [23] | UK | Outpatient | *n* = 80 (85% female) | 14.7 (*SD* = ns.) | 73% BPD, 27% no. ns. | MBT-A | TAU | 12 months |
| Mehlum et al., 2014 [24] | Norway | Outpatient | *n* = 77 (88.3% female) | 15.6 (*SD* = 1.5) | 20.5% BPD, 79.5% ≥ 2 + self-harm | DBT-A | EUC | 19 weeks |
| Salzer et al., 2014 [25] | Germany | Inpatient (experimental), Outpatient (control) | *n* = 39 (female ns.) | ns. | 100% BPD | PiM | WL/ TAU | ~ 6 months |
| Santisteban et al., 2015 [29] | US | Outpatient | n = 40 (37.5% female) | 15.8 (*SD* = 0.8) | 100% BPD | I-BAFT | IDC | 7 months |
| McCauley et al., 2018 [26] | US | Outpatient | *n* = 173 (94.8% female) | 14.89 (*SD* = 1.47) | 53.2% BPD, 46.8% ≥ 3 | DBT-A | IGST | 6 months |
| Beck et al., 2020 [20] | Denmark | Outpatient | *n* = 112 (99.1% female) | 15.8 (*SD* = 1.1) | 96% BPD, 4% ≥ 4 | MBT-G | TAU | 12 months |

*Note.* CAT = Cognitive analytic therapy; ERT = Emotion regulation training; SFET = Specialist first episode psychosis treatment; MBT-A = Mentalization-based treatment for adolescents; DBT-A = Dialectical behavior therapy for adolescents; PiM = Psychoanalytic-interactional method; I-BAFT = Integrative borderline personality disorder-oriented adolescent family therapy; MBT-G = Mentalization-based treatment in groups; SGCC = Standardized good clinical care; TAU = Treatment as usual; EUC = Enhanced usual care; WL = Waiting list; IDC = Individual drug counseling; IGST = Individual and group supportive therapy.

**Table 2. Overview of the measures used in this review to assess primary and secondary outcomes.**

| Measure | Number of trials |
|---|---|
| **BPD severity** | |
| The Borderline Personality Features Scale for Children (BPFS-C) [30] | 2 |
| The Structured Clinical Interview for DSM-IV Axis II Disorders (SCID-II) [32] | 1 |
| Borderline Symptom List (BSL) [33] | 1 |
| The Millon Adolescent Clinical Inventory (MACI) [34] | 1 |
| The Borderline Personality Disorder Severity Index-IV (BDSI-IV) [35,36] | 1 |
| **Psychopathological syndromes** | |
| The Brief Psychiatric Rating Scale (BPRS) [37] | 1 |
| The Montgomery-Åsberg Depression Rating Scale (MADRS) [38,39] | 1 |
| The Short Mood and Feelings Questionnaire (MFQ) [40] | 1 |
| The Beck's Depression Inventory for Youth (BDI-Y) [41] | 1 |
| The Symptoms Checklist-90-Revised (SCL-90-R) [42] | 1 |
| **Impulsivity** | |
| The Risk-Taking and Self-Harm Inventory (RTSHI) [43] | 2 |
| The Overt Aggression Scale-Modified for outpatients (OAS-M) [44] | 1 |
| The Youth Self-Report (YSR) [45] | 1 |
| The Suicide Attempt Self-Injury Interview (SASII) [46] | 1 |
| Self-developed semi-structured interview [22] | 1 |
| Self-reported self-harm episodes [24] | 1 |
| **Alcohol or substance abuse** | |
| The Substance Dependence Scale (SDS) [47] | 1 |
| **Functioning** | |
| The Social and Occupational Functioning Assessment Scale (SOFAS) [48] | 2 |
| The Children's Global Assessment Scale (CGAS) [49] | 2 |
| The Global Assessment of Functioning (GAF) [12] | 1 |
| **Quality of life** | |
| The Youth Quality of Life—Research Version (YQOL) [50] | 1 |

## Measures

For an overview of the measures used to assess the primary and secondary outcomes in this review please see Tables 2 and 3.

## Risk of bias assessment in included trials

**Generation of allocation sequence.**   We found low risk of bias in the generation of allocation sequence in nine of the included trials. Here the allocation was assigned by block randomization, computer-generated number-list, minimization, computerized adaptive minimization, or by drawing lots [20,22–24,26–29,31]. In one trial we found an unclear risk of bias due to insufficient information as to what simple randomization entailed [25].

**Allocation concealment.**   The risk of bias in allocation concealment was low in six trials since the randomization procedure was either concealed from the therapists or investigators by use of computer programs, independent blinded personnel, separate envelopes or external groups [20,22–24,26,27]. The risk of bias was high in one trial due to lack of blinding [28], and in three trials no information on allocation concealment was provided thus warranting an unclear risk of bias [25,29,31].

**Blinding.**   Due to the fact that it is difficult to blind participants and personnel in psychotherapeutic trials, all trials had a high risk of bias in this category. It is, however, possible to

**Table 3. Measures used within the included trials.**

| | BPD severity | Psychopathological syndromes | Impulsivity | Alcohol or substance abuse | Functioning | Quality of life |
|---|---|---|---|---|---|---|
| Chanen et al. [22] | SCID-II | - | Self-developed semi-structured interview | - | SOFAS | - |
| Schuppert et al. [31] | BDSI-IV | - | YSR | - | - | - |
| Schuppert et al. [27] | BDSI-IV | SCL-90-R | - | - | - | YQOL |
| Gleeson et al. [28] | - | BPRS | OAS-M | SDS | SOFAS | - |
| Rossouw & Fonagy [23] | BPFS-C | MFQ | RTSHI | - | - | - |
| Mehlum et al. [24] | BSL | MADRS | Self-reported self-harm episodes | - | CGAS | - |
| Salzer et al. [25] | - | - | - | - | GAF | - |
| Santisteban et al. [29] | MACI | - | - | - | - | - |
| McCauley et al. [26] | - | - | SASII | - | - | - |
| Beck et al. [20] | BPFS-C | BDI-Y | RTSHI | - | CGAS | - |

blind assessments to avoid detection bias. In nine of the trials, assessments were conducted by a blinded assessor, thus low risk of bias [20,22–28,31]. In one trial we considered the risk of bias high since assessors were not blind to the intervention [19].

**Incomplete outcome data.** Three trials had low risk of bias due to low attrition rates and use of intention-to-treat analysis [24–26]. Four trials had unclear risk of bias either due to high attrition rates, uneven attrition rates in the intervention groups, unclear attrition rates or no reasons for attrition stated which makes it unclear whether reasons could be unevenly distributed across groups [20,22,27,29]. Three trials had high risk of bias due to high attrition rates, no reasons for attrition reported, no report of imputation method or intention-to-treat or if only completers were included in the analysis [23,28,31].

**Selective reporting.** Six trials had published protocols before the trial [20,23–24,26–28]. However, the outcomes mentioned in the protocols differed to some extent from those in the trial publications in five trials, thus warranting a high risk of bias [20,23,26–28]. Four of the trials did not publish protocols [22,25,29,31] or there were uncertainties around post hoc analyses not mentioned in the protocol [24], and therefore the risk of bias was considered unclear.

**Other potential sources of bias.** Other potential sources of bias included treatment adherence bias, allegiance bias, and attention bias. Only one trial had no other potential sources of bias [22]. With regards to treatment adherence bias, five trials were considered to have low risk of bias [20,22,26–29], and five trials had an unclear risk of bias since adherence ratings either were not applied or not reported [23–25,29,31]. Three trials were considered to have low risk of allegiance bias [20,22,28], and seven trials a high risk of allegiance bias since the developers of the experimental treatment were authors or the first author was trained in the experimental treatment [23–27,29,31]. With regards to attention bias, four trials were considered to have low risk of bias due to matched treatment dosage of the interventions [22,23,26,29], and three trials were considered to have a high risk of bias since the experimental treatment entailed more attention [20,24,25]. Three trials did not give sufficient information to determine whether there was attention bias, thus warranting an unclear risk of bias [27,29,31].

To summarize, all of the 10 included trials were at high risk of bias according to *the Cochrane Handbook for Systematic Reviews of Interventions* [16] (Figs 2 and 3). Only one trial had no high risk of bias on any singular domain [22], but since there was an unclear risk of bias on two domains, it was also considered high risk of bias.

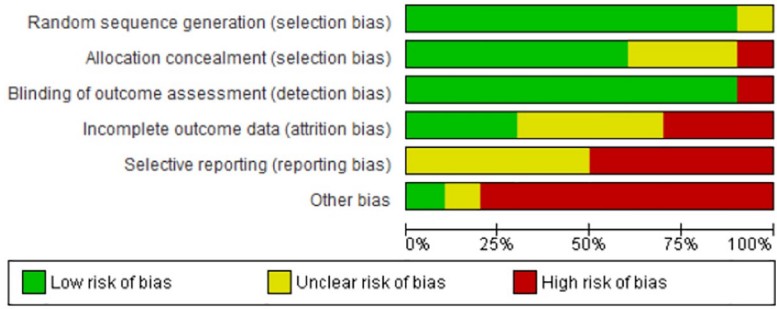

**Fig 2. Graph over the risk of bias in the included trials.**

## Effects of the interventions

Results from our primary outcome and secondary outcomes are presented below. The effect sizes are presented as MD, and if necessary, SMD. Two authors were contacted with regards to

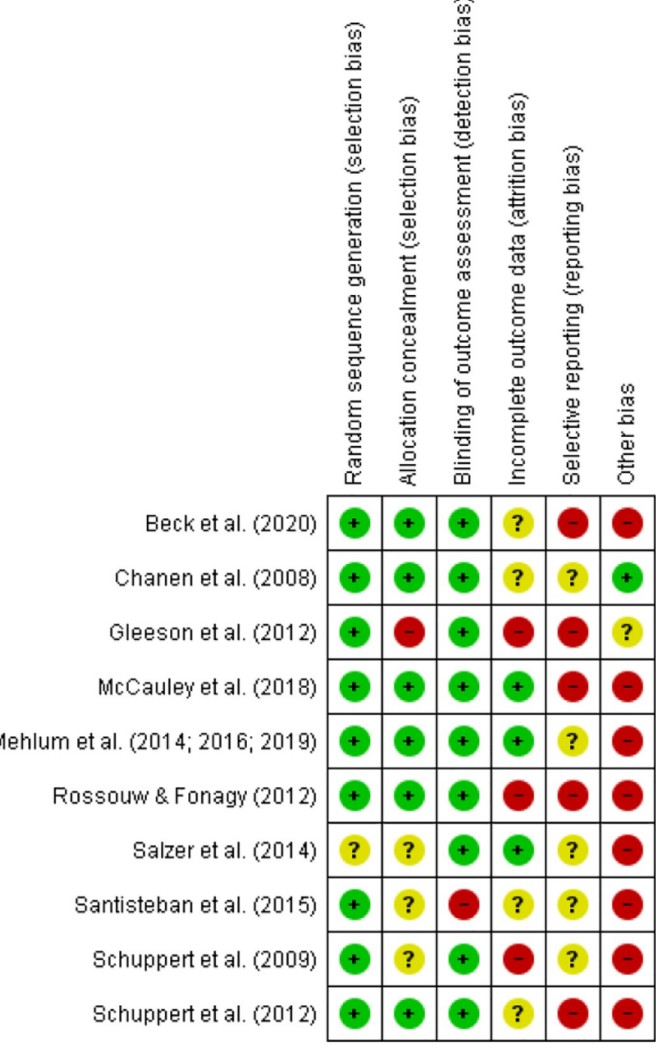

**Fig 3. Summary of the risk of bias in the included trials.**

delivering missing data or for clarifying reasons. We received information back from both authors [8,51].

**Primary outcome on BPD severity.** Seven trials included measures of BPD severity [20,22–24,27,29,31].

*Meta-analyses*: Two studies of ERT [27,28] and two studies of MBT—one with individual MBT treatment [23] and one with group-based MBT treatment [20] reported on this outcome. We found no evidence of an effect of neither ERT (MD = -2.24, 95% CI = -5.46 to 0.97, two trials, 128 participants, $I^2$ = 0%, Z = 1.37, P = 0.17), nor MBT (MD = -3.06, 95% CI = -9.51 to 3.39, two trials, 191 participants, $I^2$ = 55%, Z = 0.93, P = 0.35).

*Single study results*: Single study results included CAT [22], DBT-A [24,52,53], I-BAFT [29], and MBT-G [21].

*CAT*: Chanen et al. [22] measured BPD severity after EOT, and at six and 18 months follow-up, respectively. They found no significant differences between CAT and MGCC on the SCID-II at EOT (MD = -0.75, 95% CI = -2.37 to 0.87, 69 participants, Z = 0.91; P = 0.37), at 6 months follow up (MD = -0.41, 95% CI = -2.23 to 1.41, 70 participants, Z = 0.44; P = 0.66), nor at 18 months follow up (MD = 0.30, 95% CI = -1.99 to 2.59, 68 participants, Z = 0.26; P = 0.80).

*DBT-A*: Mehlum et al. [24,52,53] measured BPD severity after EOT, and at one and three years follow-up, respectively. They found a significant difference between DBT-A and EUC on the BSL at EOT (MD = -13.41, 95% CI = -21.77 to -5.05, 77 participants, Z = 3.14; P = 0.002), but no significant differences at one year follow-up (MD = 1.36, 95% CI = -7.73 to 10.45, 75 participants, Z = 0.29; P = 0.77) nor at three years follow-up (MD = -2.33, 95% CI = -12.20 to 7.54, 71 participants, Z = 0.46; P = 0.64).

*I-BAFT*: Santisteban et al. [29] measured BPD severity after EOT and found no difference between I-BAFT and IDC on the MACI (OR = 2.00, 95% CI = 0.52 to 7.72, 40 participants, Z = 1.01; P = 0.31).

*MBT-G (follow-up data)*: Jørgensen et al. [21] followed up on participants from Beck et al. [20] 3 and 12 months after EOT. They found no significant differences on BPD severity between MBT-G and TAU on the BPFS-C at 3 months follow up (MD = 0.70, 95% CI = -5.52 to 6.92, 93 participants, Z = 0.22; P = 0.83), nor at 12 months follow up (MD = 0.90, 95% CI = -5.04 to 6.84, 97 participants, Z = 0.30; P = 0.77).

## Secondary outcomes

**Psychopathological syndromes.** Five trials included measures of psychopathological syndromes [20,23,24,27,28].

*Meta-analyses*: It was only possible to combine data on psychopathological syndromes from two trials on MBT [20,23] into a meta-analysis. Rossouw & Fonagy [23] measured depression using the SMFQ, and Beck et al. [20] measured depression using the BDI-Y. We found no evidence of an effect of MBT compared with TAU on depression (SMD = -0.88, 95% CI = -2.81 to 1.04, two trials, 164 participants, $I^2$ = 97%, Z = 0.90, P = 0.37).

*Single study results*: single study results included CAT [28], DBT-A [24,52,53], ERT [27], and MBT-G [21].

*CAT*: Gleeson et al. [28] measured psychotic symptoms at EOT and six months follow-up. They found no significant differences between CAT plus SFET compared with SFET on the BPRS between the interventions at EOT, (MD = -6.10, 95% CI = -15.01 to 2.81, 9 participants, Z = 1.34; P = 0.18), nor at six months follow-up (MD = -11.70, 95% CI = -25.11 to 1.71, 8 participants, Z = 1.71; P = 0.09).

*DBT-A*: Mehlum et al. [24,52,53] measured depression at EOT, and at one and three year follow-up, respectively. They did not find significant differences between DBT-A and EUC on the MADRS at EOT (MD = -3.47, 95% CI = -6.97 to 0.03], 77 participants, Z = 1.94; P = 0.05), at one year follow-up (MD = -0.64, 95% CI = -4.53 to 3.25, 75 participants, Z = 0.32; P = 0.75) nor at three years follow-up (MD 1.36, 95% CI = -1.96 to 4.68, 71 participants, Z = 0.80; P = 0.42).

*ERT*: Schuppert [27] measured general psychological complaints with the total score of the SCL-90-R, and found no significant difference between ERT plus TAU compared with TAU at EOT (MD = -2.81, 95% CI = -30.33 to 24.71, 96 participants, Z = 0.20; P = 0.84).

*MBT-G (follow-up data)*: Jørgensen et al. [21] followed up on participants from Beck et al. [20] 3 and 12 months after EOT. They found no significant differences in depression between MBT-G and TAU on the BDI-Y at 3 months follow up (MD = 4.30, 95% CI = -0.63 to 9.23, 93 participants, Z = 1.71; P = 0.09), nor at 12 months follow up (MD = 1.30, 95% CI = -2.90 to 5.50, 97 participants, Z = 0.61; P = 0.54).

**Impulsivity.**   Seven trials included measures of impulsivity [20,22–24,26,28,31].

*Meta-analyses*: Two trials on MBT [20,23] measured self-harm using the same measure RTSHI, and we therefore combined the results from EOT into a meta-analysis. Since the results were presented as dichotomous data in Rossouw & Fonagy [23] and as continuous data in Beck et al. [20], an inverse variance method was applied. We found no significant effect of MBT compared with TAU on reduced self-harm (OR = 0.61, 95% CI = 0.13 to 2.91, two trials, 155 participants, $I^2$ = 82%; P = 0.53).

Two trials on DBT-A [24,26] measured self-harm, and we were therefore able to combine their results from EOT into a meta-analysis. Since the time of follow-up differed between the two trials (six months compared to one and three years), only EOT data were combined in a meta-analysis. Mehlum et al. [24] used self-reported self-harm episodes, and McCauley et al. [26] used the SASII. Since the results were presented as dichotomous data in McCauley et al. [26] and as continuous data in Mehlum et al. [24], an inverse variance method was applied. We found evidence of an effect of DBT-A compared with control interventions on self-harm at EOT (OR = 0.45, 95% CI = 0.26 to 0.76, two trials, 212 participants, $I^2$ = 0%; P = 0.003).

Two trials on CAT [22,28] measured impulsivity (self-harm incidents and suicidality), and we therefore combined the results from EOT and at six months follow-up into a meta-analysis. We found no significant effect of CAT over control interventions on impulsivity at EOT (SMD = 0.04, 95% CI = -0.40 to 0.49, two trials, 78 participants, $I^2$ = 0%; P = 0.85) or at six months follow-up (SMD = 0.08, 95% CI = -0.36 to 0.53, two trials, 78 participants, $I^2$ = 0%; P = 0.72).

*Single study results*: single study results included CAT [22], DBT-A [24,26,52,53], ERT [31], and MBT-G [21].

*CAT (follow-up data)*: Chanen et al. [22] measured self-harm incidents (including suicide attempts and NSSI) at EOT and at follow-up after 6 months and 18 months (the first two time points are presented above in a meta-analysis). Comparable to the results from the meta-analysis, they found no differences between CAT and MGCC at 18 months follow-up (MD = 10.04, 95% CI = -3.56 to 23.64, 68 participants, Z = 1.45; P = 0.15).

*DBT-A (follow-up data)*: Mehlum et al. [24,52,53] measured frequency of self-harm episodes at EOT, and at one and three years follow-up, respectively (the first two time points are presented above in a meta-analysis). They found a difference between DBT-A and EUC on self-reported self-harm episodes at one year follow-up (MD = -9.30, 95% CI = -17.43 to -1.17, 75 participants, Z = 2.24; P = 0.02), but not at three years follow-up (MD = -12.62, 95% CI = -27.53 to 2.29], 71 participants, Z = 1.66; P = 0.10).

*DBT-A (follow-up data)*: McCauley et al. [26] measured rates self-harm episodes on the SASII at EOT and six months follow-up (the first time point is presented above in a meta-analysis). Self-harm was categorized into four groups according to occurrence (0, 1–3, 4–9, and ≥10). At six months follow-up, there was no significant difference between DBT-A and IGST in the occurrence of 0 self-harm episodes between the interventions (OR = 1.75, 95% CI = 0.86 to 3.53, 129 participants, Z = 1.55; P = 0.12).

*ERT*: Schuppert et al. [31] measured externalizing symptoms with the YSR at EOT and found no difference between ERT plus TAU compared with TAU at EOT (MD = -1.40, 95% CI = -9.36 to 6.56, 26 participants, Z = 0.34; P = 0.73).

*MBT-G (follow-up data)*: Jørgensen et al. [21] followed up on participants from Beck et al. [20] 3 and 12 months after EOT. They found no significant differences between MBT-G and TAU on self-harm on the RTSHI at 3 months follow up (MD = 0.20, 95% CI = -4.44 to 4.84, 93 participants, Z = 0.08; P = 0.93), nor at 12 months follow up (MD = 0.70, 95% CI = -3.90 to 5.30, 97 participants, Z = 0.30; P = 0.77).

**Alcohol or substance abuse.**   Only one trial included measures of alcohol or substance abuse [28], therefore no meta-analysis was possible. Single study results included one trial on CAT.

*CAT*: Gleeson et al. [28] measured substance dependence at EOT and six months follow-up. They found no differences between CAT plus SFET compared with SFET on the SDS at EOT (MD = 3.00, 95% CI = -3.63 to 9.63, 9 participants, Z = 0.89; P = 0.38), nor at six months follow-up (MD = -0.80, 95% CI = -6.10 to 4.50, 8 participants, Z = 0.30; P = 0.77)

**Functioning.**   Five trials included measures of functioning [20,22,24,25,28].

*Meta-analyses*: We combined data from two trials on CAT [22,28] in a primary meta-analysis of functioning at EOT and at six months follow-up. We found no evidence of an effect of CAT over control interventions at EOT (MD = 7.54, 95% CI = -5.88 to 20.96, two trials, 78 participants, $I^2$ = 68%; Z = 1.10, P = 0.27), nor at six months follow-up (MD = 6.15, 95% CI = -9.37 to 21.67, two trials, 78 participants, $I^2$ = 75%; Z = 0.78, P = 0.44).

*Single study results*: single study results included CAT [22], DBT-A [24,52,53], MBT-G [20,21], and PiM [25].

*CAT (follow-up data)*: Chanen et al. [22] measured functioning at EOT, and six and 18 months follow-up (the first two time points are presented above in a meta-analysis). They found no differences between CAT and MGCC on the SOFAS at 18 months follow-up (MD = -3.56, 95% CI = -9.22 to 2.10], 68 participants, Z = 1.23; P = 0.22).

*DBT-A*: Mehlum et al. [24,52,53] measured global level of functioning at EOT, and one and three year follow-up, respectively. They found no differences between DBT-A and EUC on the CGAS at EOT (MD = -0.01, 95% CI = -5.91 to 5.17, 75 participants, Z = 0.00; P = 1.00), at one year follow-up (MD = 1.46, 95% CI = -4.44 to 7.36, 75 participants, Z = 0.48; P = 0.63), nor at three years follow-up (MD = -1.15, 95% CI = -6.49 to 4.19, 71 participants, Z = 0.42; P = 0.67).

*MBT-G*: Beck et al. [20] measured global level of functioning at EOT and found no difference on the CGAS between MBT-G and TAU (MD = -0.60, 95% CI = -6.29 to 5.09, 84 participants, Z = 0.21; P = 0.84). Jørgensen et al. [21] followed up on participants from Beck et al. [20] 3 and 12 months after EOT. They likewise found no significant differences between MBT-G and TAU on the CGAS at 3 months follow up (MD = -0.50, 95% CI = -5.21 to 4.21, 93 participants, Z = 0.21; P = 0.84), nor at 12 months follow up (MD = -0.30, 95% CI = -5.57 to 4.97, 91 participants, Z = 0.11; P = 0.91).

*PiM*: Salzer [25] measured global level of functioning at EOT and found a difference between PiM and WL/TAU on the GAF (MD = 13.18, 95% CI = 7.70 to 18.66, 39 participants, Z = 4.72; P = 0.00001).

**Table 4. Attrition rates in the included trials.**

| Trial | Experimental interventionn/N (%) | Control interventionn/N (%) | Relative effect (95% CI) | P |
|---|---|---|---|---|
| Chanen et al., 2008[a] [22] | CAT = 12/44 (27) | SGCC = 11/42 (26) | OR 1.06, 95% CI 0.41 to 2.75 | .91 |
| Schuppert et al., 2009[b] [31] | ERT + TAU = 9/23 (39) | TAU = 3/20 (15) | OR 3.64, 95% CI 0.82 to 16.10 | .09 |
| Schuppert et al., 2012[c] [27] | ERT + TAU = 9/54 (17) | TAU = ns. | - | - |
| Gleeson et al., 2012[d] [28] | CAT + SFET = 6/8 (75) | SFET = ns. | - | - |
| Rossouw & Fonagy, 2012 [23] | MBT-A = 20/40 (50) | TAU = 23/40 (58) | OR 0.74, 95% CI 0.31 to 1.78 | .50 |
| Mehlum et al., 2014[c] [24] | DBT-A = 10/39 (26) | EUC = 11/38 (29) | OR 0.85, 95% CI 0.31 to 2.31 | .74 |
| Salzer et al., 2014 [25] | PiM = ns. | WL/TAU = ns. | - | - |
| Santisteban et al., 2015 [29] | I-BAFT = 3/20 (15) | IDC = 2/20 (10) | OR 1.59, 95% CI 0.24 to 10.70 | .63 |
| McCauley et al., 2018 [26] | DBT-A = 20/86 (23) | IGST = 39/87 (45) | OR 0.37, 95% CI 0.19 to 0.72 | **.003** |
| Beck et al., 2020 [20] | MBT-G = 32/56 (57) | TAU = 14/56 (25) | OR 4.00, 95% CI 1.79 to 8.94 | **.0007** |

[a] = not including patients who negotiated early termination due to enough treatment received

[b] = lost to second assessment

[c] = attended less than half of the sessions

[d] = completed less than 16 sessions

**Quality of life.** Only one trial measured quality of life [27], therefore no meta-analysis was possible. Single study results included one trial on ERT [27].

*ERT*: Schuppert et al. [27] measured quality of life at EOT and found no difference on the YQOL between ERT plus TAU and TAU (MD = -0.21, 95% CI = -5.48 to 5.06, 97 participants, Z = 0.08; P = 0.94).

**Adverse events.** In the majority of the trials, no harmful effects of the interventions were mentioned. In one trial, one participant in the control intervention died by suicide [26]. In another trial, one participant in the experimental intervention deteriorated on positive psychotic symptoms [28].

**Attrition.** Attrition rates in the experimental interventions varied from 15 to 75%, and from 10 to 57.5% in the control interventions. We could not retrieve information on drop-out in the experimental intervention in one trial [25], and we could not retrieve information on drop-out in the control intervention in three trials [25,27,28]. Please see Table 4 for information on attrition rates and the relative effect between the experimental and control interventions in the included trials.

## Trial Sequential Analysis and quality of the evidence (GRADE)

We did a Trial Sequential Analysis on our primary outcome in cases where meta-analyses were possible (Figs 4 and 5), thus comprising four trials on the primary outcome BPD severity (two on ERT, and two on MBT). For ERT, we could not find a definition of minimum clinical relevance. Therefore, we calculated this on the basis of a half standard deviation which was 4.6 [54]. The TSA adjusted CI was -5.46 to 0.98) and the RIS was 128 participants, and the total number of participants was 128. The cumulated Z curve enters the futility area, and any anticipated intervention effect can be rejected at this point. Please see Table 5 for summary of findings including quality of the evidence.

For MBT, the minimum clinical relevance on the BPFS-C was 12 units [20]. The TSA adjusted CI was -11.05 to 4.93) and the RIS was 113 participants, and the total number of participants was 191. The cumulated Z curve enters the futility area, and any anticipated intervention effect can be rejected at this point.

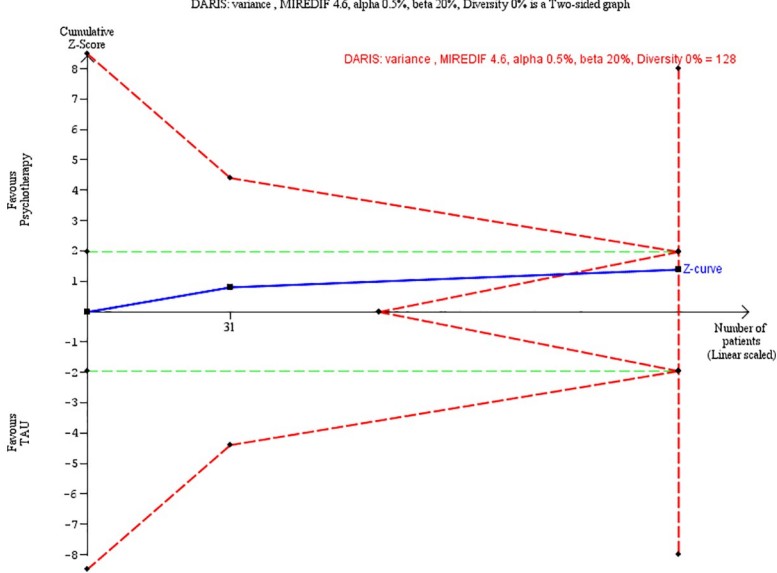

**Fig 4. Trial Sequential Analysis for ERT.**

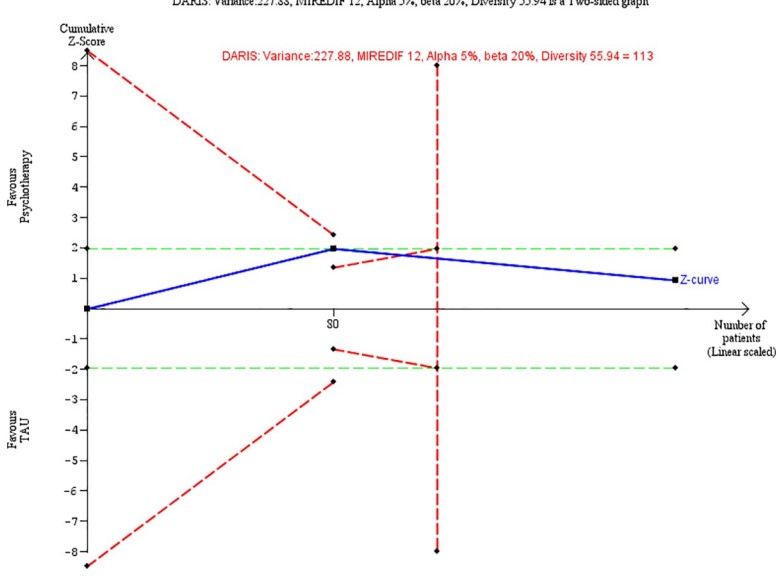

**Fig 5. Trial Sequential Analysis for MBT.**

## Discussion

We conducted this systematic review to examine the effectiveness of psychological therapies for adolescents with BPD or BPD features. We assessed 563 full-text papers and included 10 trials consisting of 775 adolescents with BPD or BPD features for this review. To our knowledge this is the most comprehensive systematic review and meta-analysis on psychological therapies for adolescents with BPD or BPD features.

**Table 5. Summary of findings table (EOT data).**

| Outcomes | No. of participants | Quality of the evidence (GRADE) | Relative effect (95% CI) | P |
|---|---|---|---|---|
| **Borderline severity (primary)** | | | | |
| ERT | 128 (2 trials) | ⊕⊝⊝⊝ very low [a, b] | MD -2.24, 95% CI -5.46 to 0.97 | .17 |
| MBT | 191 (2 trials) | ⊕⊝⊝⊝ very low [a, b, c] | MD -3.06, 95% CI -9.51 to 3.39 | .35 |
| CAT | 69 (1 trial) | ⊕⊝⊝⊝ very low [a, b, c] | MD -0.75, 95% CI -2.37 to 0.87 | .37 |
| DBT-A | 77 (1 trial) | ⊕⊝⊝⊝ very low [a, b, c] | MD -13.41, 95% CI -21.77 to -5.05 | **.002** |
| I-BAFT | 40 (1 trial) | ⊕⊝⊝⊝ very low [a, b, c] | OR 2.00, 95% CI 0.52 to 7.72 | .31 |
| **Psychopathological syndromes** | | | | |
| MBT | 164 (2 trials) | ⊕⊝⊝⊝ very low [a, b, c] | SMD -0.88, 95% CI -2.81 to 1.04 | .37 |
| CAT | 9 (1 trial) | ⊕⊝⊝⊝ very low [a, b, c] | MD -6.10, 95% CI -15.01 to 2.81 | .18 |
| ERT | 96 (1 trial) | ⊕⊝⊝⊝ very low [a, b, c] | MD -2,81, 95% CI -30.33 to 24.71 | .84 |
| DBT-A | 77 (1 trial) | ⊕⊝⊝⊝ very low [a, b, c] | MD -3.47, 95% CI -6.97 to 0.03 | .05 |
| **Impulsivity** | | | | |
| MBT | 155 (2 trials) | ⊕⊝⊝⊝ very low [a, b, c] | OR 0.61, 95% CI 0.13 to 2.91 | .53 |
| DBT-A | 212 (2 trials) | ⊕⊝⊝⊝ very low [a, b] | OR 0.45, 95% CI 0.26 to 0.76 | **.003** |
| CAT | 78 (2 trials) | ⊕⊝⊝⊝ very low [a, b, c] | SMD 0.04, 95% CI -0.40 to 0.49 | .85 |
| ERT | 26 (1 trial) | ⊕⊝⊝⊝ very low [a, b, c] | MD -1.40, 95% CI -9.36 to 6.56 | .73 |
| **Substance or alcohol abuse** | | | | |
| CAT | 9 (1 trials) | ⊕⊝⊝⊝ very low [a, b, c] | MD 3.00, 95% CI -3.63 to 9.63 | .77 |
| **Functioning** | | | | |
| CAT | 78 (2 trials) | ⊕⊝⊝⊝ very low [a, b, c] | MD 7.54, 95% CI -5.88 to 20.96 | .27 |
| MBT | 84 (1 trial) | ⊕⊝⊝⊝ very low [a, b, c] | MD -0.60, 95% CI -6.29 to 5.09 | .84 |
| DBT-A | 75 (1 trial) | ⊕⊝⊝⊝ very low [a, b, c] | MD -0.01, 95% CI -5.19 to 5.17 | 1.00 |
| PiM | 39 (1 trial) | ⊕⊝⊝⊝ very low [a, b, c] | MD 13.18, 95% CI 7.70 to 18.66 | **.00001** |
| **Quality of life** | | | | |
| ERT | 97 (1 trial) | ⊕⊝⊝⊝ very low [a, b, c] | MD -0.21, 95% CI -5.48 to 5.06 | .94 |

GRADE Working Group grades of evidence
**High quality**: Further research is very unlikely to change our confidence in the estimate of effect.
**Moderate quality**: Further research is likely to have an important impact on our confidence in the estimate of effect and may change the estimate.
**Low quality**: Further research is very likely to have an important impact on our confidence in the estimate of effect and is likely to change the estimate.
**Very low quality**: We are very uncertain about the estimate.

[a] Downgraded due to risk of bias on more than one domain

[b] Downgraded due to imprecision (either based on 1 trial or few patients; wide CI)

[c] Downgraded due to inconsistency (high heterogeneity)

The duration of the trials ranged from 19 weeks [24] to 12 months [20,23]. Most were conducted in outpatient clinics in the US, Europe, or Australia. Nine of the trials had at least one category with high risk of bias, and one had no high risk of bias but two categories with unclear risk of bias [22]. In total, we thus assessed all trials as having high risk of bias, which could lead to systematic errors, i.e. overestimating benefits and underestimating harms. These findings do not equate to wrong conclusions with regards to the findings in the included studies or that psychological therapies for adolescents with BPD or BPD features do not work, but reviews such as these are needed to support and inform clinical practice with regards to the evidence base.

In accordance with *the Cochrane Handbook for Systematic Reviews of Interventions* [16], we performed meta-analyses on our primary outcome BPD severity at EOT. On the primary outcome, no statistical significant differences were found between the experimental interventions or control interventions in our two meta-analyses of MBT and ERT. TSAs showed that the

cumulated Z curve entered the futility area, and therefore any anticipated intervention effect can be rejected at this point of time. Single study results on the primary outcome BPD severity included a statistical significant difference between DBT and EUC at EOT (but not in the follow-up period) [24,52,53], and no statistically significant differences between the experimental and control interventions in the remaining trials [22,29].

For the secondary outcomes, we only found two instances where the experimental intervention was statistically significant different from control interventions: 1) DBT-A was superior at EOT in reducing self-harm. However, this superiority vanished at six months follow-up in McCauley et al.'s trial [26], and after three years in Mehlum et al.'s trial [53], and 2) a single trial of PiM was superior to WL/TAU in increasing global level of functioning [25].

## Implications for early intervention for BPD

A recent systematic review with meta-analysis on seven trials of psychotherapies for adolescents with subclinical and BPD concluded that psychotherapies for adolescents with BPD pathology are effective for BPD-specific symptomatology, externalizing and internalizing symptoms, particularly in the short term, and also reduce the frequency of NSSI [55]. Furthermore, they concluded that the risk of bias in the included studies was generally low, and the studies were rated as being of very high quality [55]. These conclusions were based on pooled results from all experimental treatments compared to all control interventions regardless of theoretical orientation and length. There are, however, major limitations to this review. First, pooling of all experimental interventions should only be done when subgroup analysis of clinical heterogeneity can be conducted. Since there are less than ten trials within this field, this pooling of experimental interventions versus controls should be avoided [16]. Furthermore, there was no published protocol, some trials were left out, there was no use of tools to rate quality, there were overly optimistic risk of bias ratings, and lastly also issues regarding clinical heterogeneity [56]. These limitations can mislead clinicians and researchers with regards to the evidence base. Therefore, a systematic review that thoroughly addressed these limitations was warranted. We believe this review addresses the before mentioned limitations, which led to considerable different conclusions regarding the effectiveness of the included studies, the risk of bias assessments as well as ratings of the quality of the evidence.

Diagnosing BPD in youth has been controversial, and therefore there is a scarcity of RCTs on psychological therapies for this age group with BPD. For that reason, it is of uttermost importance to outline the quality of the evidence and the risk of bias in a transparent fashion. BPD is a severe mental disorder associated with enduring difficulties in achieving functional remission (especially vocational recovery), and this constitutes a costly feature of BPD that is remarkable stable without targeted intervention [4,6,57,58]. In a Danish register based study of 67,075 individuals diagnosed with BPD, the BPD group had 32% lower odds of being in work/under education after nine years as well as more impairment in long-term vocational outcome than other PDs, as well as lower labor-market attachment than other disorders (except for schizophrenia, schizotypal and delusional disorders and substance use disorders) [58]. Similar results have been found in a recent nation-wide study of Danish patients with early onset of BPD (<19 years), where BPD patients already at age 20 had reached a statistically significant lower educational level (including lower primary school grades), and were 22 times more likely to be unemployed compared to controls [59]. Furthermore, total health care costs were more than eight times higher in the BPD group [59]. This functional disability has been a key incentive to treat BPD in adolescence where the disorder is still in its early stages (including subthreshold presentations), and where BPD traits are more flexible and malleable [60].

The trials included in this review, show that we still have some way to go before we can identify effective components of early intervention. RCTs on BPD interventions generally focus on reducing BPD symptoms and especially "acute" symptoms of BPD (such as self-harm), despite the fact that these symptoms naturally remit in the transition period from adolescence to adulthood with a symptomatic switch towards symptoms of chronic dysphoria, interpersonal difficulties and persistent functional disability [4,61,62]. Patients with BPD have pointed to functional recovery as a priority of treatment [63], and therefore a focus on functional recovery has been proposed as the most important outcome of intervention and research within this field [6]. This focus on "acute symptoms" of BPD leaves vital questions on early intervention research unanswered: was the observed symptomatic reduction in the trials due to effectiveness of treatment or merely natural remission? Due to the scarcity of follow-up studies, we also do not know of the long-term effectiveness of treatment and whether it had an impact on functional outcomes.

## Limitations and strengths

The results of this review are mostly based on single trials results that include a relatively small number of participants, differing control interventions, and with a high degree of heterogeneity in the few pooled trials. This heterogeneity limits the ability to generalize the findings of this review. First and foremost, many of the RCTs aim at early detection and early prevention of BPD, but the included samples vary in terms of inclusion criteria. For instance, number of BPD criteria necessary for inclusion vary from two criteria up to five (diagnostic threshold). This marked difference in sample characteristics makes it difficult to differentiate between early intervention and regular BPD treatment, where the aim of treatment is to treat BPD and associated conditions and social disability. Secondly, the minimum age of the participants in the included trials varied from 12 (3 trials), to 14 (5 trials), to 15 (2 trials), whereas the maximum age varied from 17 (3 trials), to 18 (3 trials), to 19 (3 trials), and one trial included participants up to 25 [29]. We decided to include the latter trial, despite the fact that it consisted of participants that were not adolescents, because it was a small pilot trial consisting of sixteen participants with a mean age of 18.4 years. Lastly, the experimental treatments varied in content, format and length, and the control treatments varied in intensity and whether they were manualized or poorly defined and non-manualized.

All trials were assessed to be of high risk of bias thus raising concern of overestimating benefits and underestimating harm. We used GRADE to rate the quality of the evidence and the GRADE assessments led to downgrading the quality of the evidence to "very low quality" due to within-trial risk of bias, publication bias, imprecision and inconsistency. This means that we have very little confidence in the effect estimate [15].

We wanted to assess for publication biases, but since there were only 10 trials on adolescents with BPD or BPD features, we could not use funnel plots for comparisons nor perform Egger's statistical test for small-study effects [64] as recommended in the *Cochrane Handbook for Systematic Reviews of Interventions* [16]. Therefore, we cannot reject the possibility that there might be publication bias.

Another limitation is the lack of knowledge of possible adverse or iatrogenic effects of psychological therapies for adolescents with BPD. We saw varying attrition rates, but reasons for attrition were seldom stated, which leaves us undecided of possible adverse effects.

Our review has a number of strengths: it was conducted as a Cochrane review following the instructions from the Cochrane Handbook [16]. A protocol was published prior to conducting the review [7]. The literature search was systematic and comprehensive, and we contacted authors in cases of missing information. Additionally, we conducted TSA on the primary

outcome in cases where meta-analyses were possible, and found that the cumulated Z curve entered the futility area, and therefore that any anticipated intervention effect could be rejected at this point. We believe that our approach has led to the best possible gathering of relevant studies on adolescents with BPD or BPD features.

## Conclusions and future directions

In the majority of the trials, no superiority of the experimental intervention was found over control interventions on primary and secondary outcomes. Furthermore, the results of the included trials should be interpreted with caution due to high risk of bias and very low quality of evidence. The trials were characterized by high degrees of heterogeneity. In order to push this field forward, there needs to be more consensus on study designs that allow for comparisons. Given the enormous impact BPD and BPD features have, the case of adolescents with BPD or BPD features deserves more attention in order to avoid inauspicious developments. Importantly, the findings of this review do not equate to ineffectiveness of psychological treatments for this age group with BPD. In the majority of the trials, symptomatology decreased and functioning increased in the experimental arm as well as in the control arm, but it is unclear whether these improvements were caused by effectiveness of treatment, natural improvement or regression toward the mean. Effective treatments need to be developed and evaluated in high quality trials with larger sample sizes. Future trials should also include well-defined control interventions and include follow-up assessments to determine the long-term effectiveness of treatment. BPD severity was chosen as the primary outcome of this review due to the state in the field. However, this field would also benefit from a shift of focus to functional outcomes, and by including outcomes that were raised as important by patients and people with lived experience.

## Supporting information

**S1 Checklist. PRISMA checklist.**
(DOCX)

**S1 File. Search strategy for medline.**
(DOCX)

## Acknowledgments

We wish to give a special thank you to Trine Lacopiddan Kæstel for her important work in conducting the searches. Also a special thank you to all the co-authors on the review on psychological therapies for BPD for their great work: Birgit A. Völlm, Mickey T. Kongerslev, Jessica T. Mattivi, Christian P. Sales, Henriette Edemann Callesen, Signe S. Nielsen, Maja Lærke Kielsholm, Klaus Lieb, and the Cochrane Developmental, Psychosocial and Learning Problems Group for providing help and support.

## Author Contributions

**Conceptualization:** Mie Sedoc Jørgensen, Erlend Faltinsen, Adnan Todorovac.

**Data curation:** Mie Sedoc Jørgensen, Ole Jakob Storebø, Jutta M. Stoffers-Winterling, Erlend Faltinsen, Adnan Todorovac.

**Formal analysis:** Mie Sedoc Jørgensen.

**Funding acquisition:** Erik Simonsen.

**Investigation:** Mie Sedoc Jørgensen.

**Methodology:** Mie Sedoc Jørgensen, Ole Jakob Storebø, Jutta M. Stoffers-Winterling, Erik Simonsen.

**Project administration:** Mie Sedoc Jørgensen, Erik Simonsen.

**Resources:** Mie Sedoc Jørgensen.

**Software:** Mie Sedoc Jørgensen, Ole Jakob Storebø.

**Supervision:** Ole Jakob Storebø, Erik Simonsen.

**Writing – original draft:** Mie Sedoc Jørgensen.

**Writing – review & editing:** Ole Jakob Storebø, Jutta M. Stoffers-Winterling, Erlend Faltinsen, Adnan Todorovac, Erik Simonsen.

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
