## [Decision Letter · Decision Letter 0]

10 Aug 2020

PONE-D-20-16971

Psychological therapies for adolescents with Borderline Personality Disorder (BPD) or BPD features – a systematic review of randomized clinical trials with meta-analysis and Trial Sequential Analysis

PLOS ONE

Dear Dr. Jørgensen,

Congratulations on having conducted an excellent systematic review on a very important topic. As you can see, one reviewer expressed concerns about the co-publication with the recently published Cochrane review on the topic. This review also included the data on adolescents, which technically means that this submission to PlosOne is a co-publication. I would also like to be very clear that you have been transparent about this in your acknowledgements, thank you!

I have then made contact with the Editorial office of PlosOne to clarify the issue of co-publications. Unfortunately, PlosOne does indeed not have a co-publication agreement or policy but only accepts updated reviews or extensions of previous work (including Cochrane reviews). The Editorial office wrote that "they would only consider submissions of this nature if the overlap between the primary results of the submission have not been reported before. We would expect that authors outline how their work advances on their previous publication in the submitted manuscript."

Given these formal reasons, I have to unfortunately send the manuscript back to you. There are two potential options to proceed: (1) you could re-submit the paper and clearly develop an own and less overlapping focus (I do not know whether this is at all possible, and I unfortunately will need to reject the paper if this is not the case, e.g. a simple sub-group analysis seems not sufficient); or (2) you would probably need to withdraw your submission, and submit to a journal with co-publication agreement.

My apologies for not bringing better news on this occasion.

In case you would like to chose option 1, please submit your revised manuscript by Sep 05 2020 11:59PM. If you will need more time than this to complete your revisions, please reply to this message or contact the journal office at plosone@plos.org. Please include the following items when submitting your revised manuscript:

Kind regards,

Michael Kaess, M. D.

Academic Editor

PLOS ONE

Journal Requirements:

2. Please provide the full electronic search strategy for at least one database, including any limits used, such that it could be repeated.

3. Please ensure that you refer to Figures 2 and 3 in your text as, if accepted, production will need this reference to link the reader to the figure.

4. We note you have included a table to which you do not refer in the text of your manuscript. Please ensure that you refer to Tables 4 and 5 in your text; if accepted, production will need this reference to link the reader to each Table.

5. Please include captions for your Supporting Information files at the end of your manuscript, and update any in-text citations to match accordingly. Please see our Supporting Information guidelines for more information: http://journals.plos.org/plosone/s/supporting-information

Reviewers' comments:

Reviewer's Responses to Questions

**Comments to the Author**

1. Is the manuscript technically sound, and do the data support the conclusions?

Reviewer #1: Yes

2. Has the statistical analysis been performed appropriately and rigorously? 

Reviewer #1: Yes

3. Have the authors made all data underlying the findings in their manuscript fully available?

Reviewer #1: No

4. Is the manuscript presented in an intelligible fashion and written in standard English?

Reviewer #1: Yes

5. Review Comments to the Author

Reviewer #1: Thank you for the opportunity to review your paper. I contacted the PLOS One Editorial Team, concerning the co-publication of this previously published Cochrane review. Unfortunately, (1) PLOS One has no co-publication agreement with Cochrane, and (2) does not accept studies for publication that have already been published, in whole or in part, elsewhere in the peer-reviewed literature. Based on this and my exchange with the Editorial Team, I am very sorry, but I have to reject the paper. Although you present additional data, the overlap between the two manuscripts is obvious. I suggest to check on other journals with existing co-publication agreement, and resubmit your paper there.

Journals with Cochrane Co-Publication agreements

https://documentation.cochrane.org/display/EPPR/Agreements+with+journals+for+the+co-publication+of+Cochrane+Reviews

6. PLOS authors have the option to publish the peer review history of their article (what does this mean?). If published, this will include your full peer review and any attached files.

Reviewer #1: No

---

## [Author Response · Author response to Decision Letter 0]

15 Aug 2020

Comment #1

Please ensure that your manuscript meets PLOS ONE’s style requirements, including those for file naming.

Response

Thank you for pointing this out. We hope the manuscript and files now meet PLOS ONE’s style requirements

Comment #2

Please provide the full electronic search strategy for at least one database, including any limits used, such that it could be repeated.

Response

This information has now been added as supplementary information, file S1 Search strategy for Medline

Comments #3 and #4

Please ensure that you refer to Figures 2 and 3 in your text as, if accepted, production will need this reference to link the reader to the figure.

We note you have included a table to which you do not refer in the text of your manuscript. Please ensure that you refer to Tables 4 and 5 in your text; if accepted, production will need this reference to link the reader to each Table.

Response

Thank you for pointing this out. The figures and tables are now referred to in the text

Comment

Please include captions for your Supporting Information files at the end of your manuscript, and update any in-text citations to match accordingly. 

Response

Thank you, this has now been added.

---

## [Decision Letter · Decision Letter 1]

25 Nov 2020

PONE-D-20-16971R1

Psychological therapies for adolescents with Borderline Personality Disorder (BPD) or BPD features – a systematic review of randomized clinical trials with meta-analysis and Trial Sequential Analysis

PLOS ONE

Dear Dr. Jørgensen,

Thank you for submitting your manuscript to PLOS ONE. After careful consideration, we feel that it has merit but does not fully meet PLOS ONE’s publication criteria as it currently stands. Therefore, we invite you to submit a revised version of the manuscript that addresses the points raised during the review process.

We look forward to receiving your revised manuscript.

Kind regards,

Michael Kaess, M. D.

Academic Editor

PLOS ONE

Reviewers' comments:

Reviewer's Responses to Questions

**Comments to the Author**

1. If the authors have adequately addressed your comments raised in a previous round of review and you feel that this manuscript is now acceptable for publication, you may indicate that here to bypass the “Comments to the Author” section, enter your conflict of interest statement in the “Confidential to Editor” section, and submit your "Accept" recommendation.

Reviewer #2: All comments have been addressed

Reviewer #3: (No Response)

2. Is the manuscript technically sound, and do the data support the conclusions?

Reviewer #2: Yes

Reviewer #3: Partly

3. Has the statistical analysis been performed appropriately and rigorously? 

Reviewer #2: Yes

Reviewer #3: No

4. Have the authors made all data underlying the findings in their manuscript fully available?

Reviewer #2: Yes

Reviewer #3: Yes

5. Is the manuscript presented in an intelligible fashion and written in standard English?

Reviewer #2: Yes

Reviewer #3: Yes

6. Review Comments to the Author

Reviewer #2: I appreciate your work that is based on a sound methodology and focused on an area of research with high clinical significance.

One critical comment from my side: You have made extreme efforts to demonstrate potential biases of the publications included in this meta-analysis. This is very helpful to classify the results. However, I would appreciate if you would have been more critical concerning the question which trial should be included in the analysis. You have included three trials (Santisteban, Salzer and McCauley) with no structured assessment of relevant constructs, especially BPD severity (see table 3). Even if the number of trials in the field of BPD in adolescents is sparse, having conducted a RCT should not be the predominant criterion. To give an example: The trial of Salzer et al. (2014) was focused on mixed disorders of conduct and emotions (F92) according to DSM-IV, and some of the patients were retrospectively classified as suffering from BPD. No results of this trial are mentioned in your paper except GAF, and there are even no data on attrition. Why didn´t you exclude this trial from your meta-analysis?

As the inclusion of these three trials does not affect your results, I recommend publication without further revision.

Reviewer #3: The manuscript deals with a highly relevant question of the efficacy of BPD-specific psychological therapies for adolescents. The review is carried out at the highest standard possible at the moment and most welcome to inform researchers and cinicians about how to proceed in research and treatment.

There is a striking lack in suffiently powered studies and studies of at least good quality. The authors make a very convincing analysis of the flaws in the current data about this topic especially with regard to risk of bias and attrition. This is a kind of warning for future research in this area to plan for high drop-out and obviously small effect sizes if any (not very encouraging..).

Although I highly value this scientific approach, I would question some central decisions made by the authors that I have problems to follow.

First of all, it remained unclear to me, how the final data set was selected. Maybe going back and forth to the just published cochrane is confusing but also in the flow chart some confusion remains about that. The search resulted in 6 studies fulfilling criteria (flow chart) and then you added another 4 and one because of the main author.. What was the rationale on this additional inclusion process and why did those studies not appear in your search and did those also fulfill inclusion criteria (should be changed in the flow chart as it sounds as if those would not fulfill criteria).

Regarding your statistical analysis: Did you include only the subsamples of BPD patients or the whole samples once the study was included. This goes a bit back and forth in your description and thus remains unclear, e.g. the attrition rates: are they related to the full samples or only the BPD/ BPD feature patients? Some studies had very low BPD percentages.. This is also not mentioned in the results part at all and not mentioned in the limitations.

My biggest problem is the pooling of data on the basis of 2 studies and with such high heterogeneity (I2>50%). I would strongly recommend to reconsider this analytic strategy as you are comparing apples and pears here. Thus, I would highly recommend to report only on single outcomes which could be justified by the high heterogeneity of studies and also settings. MBT-A in Roussow & Fonagy compared to MBT-G in Beck is very different and to pool these together is in my eyes misleading (content-wise and from a statistical point of view).

As you are very critical with the former review that pooled data with high heterogeneity, you should apply this to your data as well.

Thus, I would recommend to point out in the summary more strongly that conclusions on the efficacy of BPD specific psychological therapies can not be derived from such highly biased, low attrition and underpowered studies.. But it would be helpful to learn if changes pre-post were present in both treatments (experimental and control) which would mean that both treatments were equally effective and not that nothing works (what it looks like now). Maybe in adolescence many things work? I just want you to reflect in the discussion a bit more on problematic conlusion that could be drawn from a false or biased reading of these results.

Thank you for this important work, I hope that my comments help the improve the manuscript.

7. PLOS authors have the option to publish the peer review history of their article (what does this mean?). If published, this will include your full peer review and any attached files.

Reviewer #2: No

Reviewer #3: **Yes: **Svenja Taubner

---

## [Author Response · Author response to Decision Letter 1]

8 Dec 2020

Thank you for reviewing our manuscript. We are pleased below to respond to each of the comments made by the two reviewers:

Reviewer #2

Comment

I appreciate your work that is based on a sound methodology and focused on an area of research with high clinical significance.

Response 

We thank the reviewer for the positive feedback on our manuscript

Comment

One critical comment from my side: You have made extreme efforts to demonstrate potential biases of the publications included in this meta-analysis. This is very helpful to classify the results. However, I would appreciate if you would have been more critical concerning the question which trial should be included in the analysis. You have included three trials (Santisteban, Salzer and McCauley) with no structured assessment of relevant constructs, especially BPD severity (see table 3). Even if the number of trials in the field of BPD in adolescents is sparse, having conducted a RCT should not be the predominant criterion. To give an example: The trial of Salzer et al. (2014) was focused on mixed disorders of conduct and emotions (F92) according to DSM-IV, and some of the patients were retrospectively classified as suffering from BPD. No results of this trial are mentioned in your paper except GAF, and there are even no data on attrition. Why didn´t you exclude this trial from your meta-analysis?

Response

We thank the reviewer for this comment. It was not our intention to include any study if it was only an RCT, as we detailed in the inclusion criteria at p. 5 (“study selection” and “types of participants”). It is not true that the three trials of Santisteban, Salzer and McCauley had “no structured assessment of relevant constructs”. As defined as an eligibility criterion, the presence of a BPD diagnosis or BPD features was established by use of structured diagnostic assessment instruments in each of the included trials (Santisteban: the Revised Diagnostic Interview for Borderlines; Salzer + McCauley: SCID) 

We agree that the non-definition of a clear cut-off number of diagnostic criteria introduces some clinical heterogeneity into the study pool. However, it reflects the situation in clinical settings where adolescents often present with substantial BPD symptoms but not a “full-blown” BPD diagnosis, but clinicians still need to react. Therefore, the broad inclusion criteria regarding participants’ characteristics (with regard to BPD presence, or BPD severity), enhance the applicability to clinical settings. 

But we agree that we could have made this more explicit in the manuscript, we have therefore added the following on page 12, lines 271-275: 

“In nine of the trials, we included the entire sample of participants, because they all met the criteria for either BPD or had BPD features [20, 22-24, 26, 28-31]. In one trial, only 59.1% of the sample had BPD (the remaining fulfilled criteria of a mixed disorder of conduct and emotions) [25], but subsample data for the BPD participants was already included into the Cochrane review [8]”

Comment

As the inclusion of these three trials does not affect your results, I recommend publication without further revision.

Response

Thank you.

Reviewer #3

Comment

The manuscript deals with a highly relevant question of the efficacy of BPD-specific psychological therapies for adolescents. The review is carried out at the highest standard possible at the moment and most welcome to inform researchers and clinicians about how to proceed in research and treatment.

There is a striking lack in suffiently powered studies and studies of at least good quality. The authors make a very convincing analysis of the flaws in the current data about this topic especially with regard to risk of bias and attrition. This is a kind of warning for future research in this area to plan for high drop-out and obviously small effect sizes if any (not very encouraging..).

Response 

We sincerely thank the reviewer for her carefully done resume of our review.

Comment

Although I highly value this scientific approach, I would question some central decisions made by the authors that I have problems to follow.

First of all, it remained unclear to me, how the final data set was selected. Maybe going back and forth to the just published cochrane is confusing but also in the flow chart some confusion remains about that. The search resulted in 6 studies fulfilling criteria (flow chart) and then you added another 4 and one because of the main author.. What was the rationale on this additional inclusion process and why did those studies not appear in your search and did those also fulfill inclusion criteria (should be changed in the flow chart as it sounds as if those would not fulfill criteria).

Response

We thank the reviewer for allowing us to clarify. We understand why it could be confusing. We did not have the same inclusion criteria in the current review as in Storebø et al. (2020) as we also added trials that included participants that presented BPD features at any level (i.e. any trial that specifically targeted BPD symptoms at a threshold or subthreshold level as an overall aim of the trial).

Six trials were included in the Cochrane review by Storebø et al. (2020), but Storebø et al. (2020) only included data on participants with full threshold BPD. When we scrutinized the references that had been excluded for the reason of patient characteristics from the Cochrane review, we identified an additional three trials that were not included because they did not deliver subsample data on BPD participants (Chanen et al., 2008; McCauley et al., 2018; Schuppert et al., 2009). We could include these three trials due to our different inclusion criteria. We were able to include yet one more trial (Beck et al., 2020) that was published after the last search had been carried out for the Cochrane review. 

This has now been made explicit on page 11, lines 254-264:

“We identified ten trials that consisted of adolescent samples with BPD or BPD features. Of these ten trials, six were included in the Cochrane review of psychological therapies for BPD, but only with data on participants who met diagnostic criteria for BPD [8]. Three trials had been excluded from the Cochrane review because either less than 70% of the total sample had full threshold BPD or because no subsample data were delivered on participants with BPD. However, these three trials all met inclusion criteria for the current review, as the re-inspection of all excludes for the reason of participant characteristics showed. One trial on mentalization-based group treatment [20-21] was published shortly after the last search had been carried out, but we were able to include this trial because MSJ was a part of the research group. The decision about trial inclusion, however, as well as risk of bias assessment and data analyses of this trial were done by other authors not affiliated with the trial (EGF and AT)” 

We have also edited the flow chart (p 12) and we hope it is clearer now.

Comment

Regarding your statistical analysis: Did you include only the subsamples of BPD patients or the whole samples once the study was included. This goes a bit back and forth in your description and thus remains unclear, e.g. the attrition rates: are they related to the full samples or only the BPD/ BPD feature patients? Some studies had very low BPD percentages.. This is also not mentioned in the results part at all and not mentioned in the limitations.

Response

Thank you for allowing us to clarify. Adolescents often present with subthreshold BPD, and had we chosen to exclude participants with subthreshold BPD, the samples would be very small. In fact, this is what was done in the Cochrane review, where only 278 adolescent participants were included compared to the 775 adolescent participants in the current review. We included all participants from nine of the trials into the analyses because the trials specifically targeted BPD pathology in all trials (meaning all participants had BPD pathology at a subthreshold or threshold level). In one case, Salzer et al., there was a subsequent publication on BPD participants only in a trial that targeted adolescents with mixed emotion and conduct disorders. Here we only published subsample data on participant with BPD, since it was not a BPD trial.

This has now been made explicit on page 12, lines 271-275:

“In nine of the trials, we included the entire sample of participants, because they all met the criteria for either BPD or had BPD features [20, 22-24, 26, 28-31]. In one trial, only 59.1% of the sample had BPD (the remaining fulfilled criteria of a mixed disorder of conduct and emotions) [25], but subsample data for the BPD participants was already included into the Cochrane review [8]”

Comment

My biggest problem is the pooling of data on the basis of 2 studies and with such high heterogeneity (I2>50%). I would strongly recommend to reconsider this analytic strategy as you are comparing apples and pears here. Thus, I would highly recommend to report only on single outcomes which could be justified by the high heterogeneity of studies and also settings. MBT-A in Roussow & Fonagy compared to MBT-G in Beck is very different and to pool these together is in my eyes misleading (content-wise and from a statistical point of view).

As you are very critical with the former review that pooled data with high heterogeneity, you should apply this to your data as well.

Response

We thank the reviewer for her recommendations. However, we respectfully disagree with the peer reviewer. Pooling the two trials is not like comparing apples and bananas. It is correct that the treatments in two trials have different formats as one is based on individual therapy and the other a group therapy, but we do not believe that this should make it inappropriate to meta-analyse the two studies. Even when the studies are using different outcome measures they can be pooled when they are measuring on the same construct. This can be done by using standardized MD (SMD). Both studies are comparing MBT versus treatment as usual. In the review by Wong, which we criticize, they pooled different types of treatment to different types of control interventions regardless of their theoretical orientation. This gives clinical heterogeneity, which should have been investigated in subgroup analyses. They did not conduct such analyses. The statistical heterogeneity in our meta-analyses should not be a reason for not conducting the meta-analyses as we followed the Grading of Recommendations, Assessment, Development, and Evaluation (GRADE) and downgraded the quality in the analyses due to the statistical heterogeneity. Furthermore, we downgraded the quality of the meta-analyses due to high risk of bias in both trials and imprecision in the meta-analyses. It is important to conduct meta-analyses whenever possible to increase statistical power (Higgins et al 2019). Both the Beck et al. and the Rossouw & Fonagy trials are underpowered due to extensive dropouts of participants in both trials. They both have risks of type 1 error due to this. Trial sequential analysis is a program that calculates the required information size for a meta-analysis, providing adjusted statistical thresholds for benefits, harms, or futility before the required information size is reached. We tested for the risks of type I and type II errors by using the Trial Sequential analysis software and we found that the required information size was reached, but as the cumulated Z curve enters the futility area, the anticipated intervention effect could be rejected.

References 

Higgins JPT, Thomas J, Chandler J, Cumpston M, Li T, Page MJ, Welch VA (editors). Cochrane Handbook for Systematic Reviews of Interventions. 2nd Edition. Chichester (UK): John Wiley & Sons, 2019.

Comment

Thus, I would recommend to point out in the summary more strongly that conclusions on the efficacy of BPD specific psychological therapies can not be derived from such highly biased, low attrition and underpowered studies..

Response

We thank the reviewer for pointing this out. We have added the following sentence to the conclusion part of the abstract:

“Due to the high risk of bias, high attrition rates and underpowered studies in this area, it is difficult to derive any conclusions on the efficacy of psychological therapies for BPD in adolescence.”

Comment 

But it would be helpful to learn if changes pre-post were present in both treatments (experimental and control) which would mean that both treatments were equally effective and not that nothing works (what it looks like now).

Maybe in adolescence many things work? I just want you to reflect in the discussion a bit more on problematic conclusion that could be drawn from a false or biased reading of these results.

Response

Thank you. Unfortunately, we cannot pool all the trials because there are too few trials and too much heterogeneity (what the Wong review did), but we agree that it is important that the conclusions of this review are not equated with no effectiveness of psychological treatments for this age group with BPD, because we do not know this. We have therefore added the following to the conclusion on page 35, lines 672-676:

“Importantly, the findings of this review do not equate to ineffectiveness of psychological treatments for this age group with BPD. In the majority of the trials, symptomatology decreased and functioning increased in the experimental arm as well as in the control arm, but it is unclear whether these improvements are caused by effectiveness of treatment, natural improvement or regression toward the mean”

Comment 

Thank you for this important work, I hope that my comments help the improve the manuscript

Response

We thank the reviewer for her positive comments and help with improving the manuscript.

Sincerely,

Mie Sedoc Jørgensen,

Corresponding author

---

## [Editor Report · Decision Letter 2]

29 Dec 2020

Psychological therapies for adolescents with Borderline Personality Disorder (BPD) or BPD features – a systematic review of randomized clinical trials with meta-analysis and Trial Sequential Analysis

PONE-D-20-16971R2

Dear Dr. Jørgensen,

We’re pleased to inform you that your manuscript has been judged scientifically suitable for publication and will be formally accepted for publication once it meets all outstanding technical requirements.

Kind regards,

Michael Kaess, M. D.

Academic Editor

PLOS ONE
---

## [Editor Report · Acceptance letter]

2 Jan 2021

PONE-D-20-16971R2 

Psychological therapies for adolescents with borderline personality disorder (BPD) or BPD features - a systematic review of randomized clinical trials with meta-analysis and Trial Sequential Analysis 

Dear Dr. Jørgensen:

I'm pleased to inform you that your manuscript has been deemed suitable for publication in PLOS ONE. Congratulations! Your manuscript is now with our production department. 

Kind regards, 

on behalf of

Prof. Dr. Michael Kaess 

Academic Editor

PLOS ONE